# Synthesis of Silica Particles Using Ultrasonic Spray Pyrolysis Method

Srecko Stopic [1,*], Felix Wenz [1], Tatjana-Volkov Husovic [2] and Bernd Friedrich [1]

1   IME Process Metallurgy and Metal Recycling, RWTH Aachen University, 52056 Aachen, Germany; felix.wenz@rwth-aachen.de (F.W.); bfriedrich@ime-aachen.de (B.F.)
2   Metallurgical Engineering Department, Faculty of Technology and Metallurgy, Karnegijeva 4, 11120 Belgrade, Serbia; tatjana@tmf.bg.ac.rs
*   Correspondence: sstopic@ime-aachen.de; Tel.: +49-176-7826-1674

**Abstract:** Silica has sparked strong interest in hydrometallurgy, catalysis, the cement industry, and paper coating. The synthesis of silica particles was performed at 900 °C using the ultrasonic spray pyrolysis (USP) method. Ideally, spherical particles are obtained in one horizontal reactor from an aerosol. The controlled synthesis of submicron particles of silica was reached by changing the concentration of precursor solution. The experimentally obtained particles were compared with theoretically calculated values of silica particles. The characterization was performed using a scanning electron microscope (SEM) and energy-dispersive X-ray spectroscopy (EDS). X-ray diffraction, frequently abbreviated as XRD, was used to analyze the structure of obtained materials. The obtained silica by ultrasonic spray pyrolysis had an amorphous structure. In comparison to other methods such as sol–gel, acidic treatment, thermal decomposition, stirred bead milling, and high-pressure carbonation, the advantage of the ultrasonic spray method for preparation of nanosized silica controlled morphology is the simplicity of setting up individual process segments and changing their configuration, one-step continuous synthesis, and the possibility of synthesizing nanoparticles from various precursors.

**Keywords:** silica; ultrasonic spray pyrolysis; synthesis

## 1. Introduction

The formation of silica from olivine in different metallurgical processes was studied very frequently in the last 50 years. Stopic [1] presented different ways for the deposition of silica in hydrometallurgical processes. As mentioned, the production of silica by the olivine route is a cheaper method than the commercial methods such as neutralization of sodium silicate solutions and flame hydrolysis because of the low cost of raw materials and the low energy requirements [2]. The produced silica has a specific surface area between 100 and 400 m$^2$/g, primary particles between 10 and 25 nm (agglomerated in clusters), and an $SiO_2$ content above 95%. Due to the high pozzolanic properties and the dispersion state, this silica powder can be applied successfully in concrete.

Mohanray et al. [3] prepared silica from corncob ash by the precipitation method. First, received corncob ash was calcined at 550 °C, 650 °C, and 750 °C for 2 h to remove the volatiles in the sample and determine the amorphous structure of silica. The thermally treated corncob ash was mixed with various concentrations of sodium hydroxide to extract pure silica using 1% of polyvinyl alcohol (PVA) as the dispersing agent. In the last step, spherical nano silica with a particle size of 25 nm was prepared from pure silica by the precipitation method. Unfortunately, starting from corn cob ash calcination and mixing with NaOH [3], precipitation with 1% polyvinyl alcohol (PVA) cannot ensure a controlled silica particle size and their purity. The impurities originate from the calcination process.

In the study of Huan et al. [4], nanosilica was synthesized by the sol–gel method from tetraethoxysilane (TEOS) with base catalysts and volumetric ratio TEOS/C2H5OH/

H2O/NH4OH: 5/30/1/1. The results showed that the prepared nanosilica were in an amorphous phase with an average size of about 60–100 nm and could be used for lead removal from waste water. Generally, the sol–gel method enables a high-purity amorphous silica powder; however, the process yields a low percentage. This method cannot ensure the formation of nonagglomerated spherical silica particles.

Powder precursors for sol–gel synthesis are very expensive, but Oi et al. [5] studied a new precursor for low-cost alternatives. A high surface area was used to form an anion surfactant sodium dodecyl sulfate, which regulates the molar concentration. The particles' size variability was changed by the precursor molar ratio of the sodium silicate solution with hydrochloric acid. The nanostructured silica particles were obtained over a range of particle sizes from 0.5 to 1 µm, with a high specific surface area and cubic and spherical shapes by changing the surfactant pH values and drying methods with the sol–gel process. An increase in pH value from 1 to 5 decreases the surface area from 858 to 630 $m^2/g$. The study of Nandanwar et al. [6] deals with the sol–gel synthesis of nanosilica, providing a basic understanding of the effect of calcination temperature on the growth of $SiO_2$ by the hydrolysis of TEOS with ethanol, deionized water, and catalyst mixture.

High-purity nanosilica was synthesized by Kim et al. [7] using acid treatment and surface modification from blast-furnace slag generated in the steel industry. Blast-furnace slag was treated with nitric acid to extract high-purity insoluble silica. Silica particles were produced using filtration and a surface modified by cation surfactant cetyltrimethyl ammonium bromide (CTAB). The size of silica particles was smallest when the modification temperature was 60 °C. The average size of silica particles modified with 3 wt.% CTAB was 107.89 nm, while the average size of unmodified silica was 240.38 nm. An acidic treatment can lead to the formation of silica gel and blockage of the whole process. The filtration is a required operation in this process. This treatment contains many operations in order to obtain silica. Due to these characteristics, we need to find other simple methods for the synthesis of very fine silica without an acidic treatment.

Synthesis of silicon dioxide nanoparticles in low-temperature atmospheric-pressure plasma was performed by Kretushev et al. [8]. Results of studies confirm that spherical nanoparticles within the range of 20–60 nm can be successfully prepared in low-temperature atmospheric-pressure plasma created with a high-frequency discharge maintained in the α mode between two plane-parallel grid electrodes. The degree of tetraethoxysilane decomposition is 80–95% and only slightly depends on the reaction parameters. Nanoparticles with a predominantly spherical shape are synthesized in the region of the high-frequency discharge.

A novel synthesis route is proposed by Stopic et al. [9] based on $CO_2$ absorption/ sequestration in an autoclave by forsterite ($Mg_2SiO_4$), which is part of the mineral group of olivines. Therefore, it is a feasible and safe method to bind carbon dioxide in carbonate compounds such as magnesite forming the spherical nanosilica at the same time, between 250 and 500 nm, as shown in Figure 1.

In contrast to sol–gel, acidic treatment, and hydrothermal synthesis using some acid and alkaline solutions, this synthesis method in an autoclave takes place in water solution at 175 °C and above 100 bar. This method is an environmentally friendly process related to the capture of carbon dioxide and the preparation of silica.

As the conventional methods for the synthesis of nanosilica from rice husk ash are energy- and time-consuming, Phoohinkong et al. [10] studied the synthesis of nanosilica from real available materials such as rice husk ash via sodium silicate solution. Nanosilica particles were obtained via alkaline extraction and a fast acid precipitation method at room temperature by adding inorganic salts and without surfactant. The flow synthesis was investigated at ambient temperature, varying the concentration of hydrochloric acid and sodium chloride, and the flow-rate while fixing the concentration of sodium silicate. The results revealed that the sodium chloride is significantly inorganic salt for the prepared nanosilica, with uniform spherical morphology (80–150 nm). In this synthesis, the silica

nanoparticles, with a diameter around 10 nm and aggregate particles of around 50 to 200 nm, were prepared.

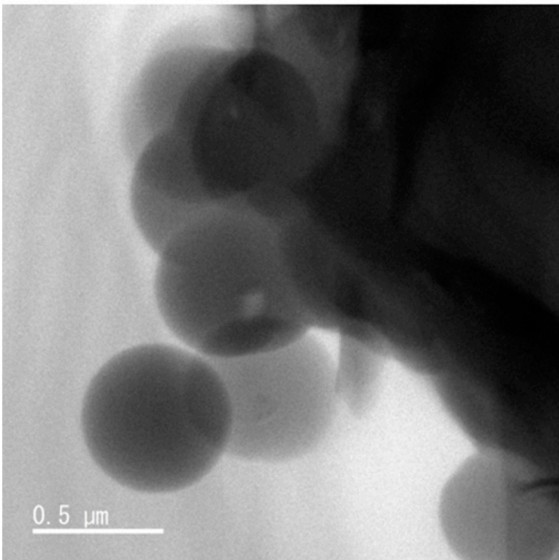

**Figure 1.** Nanosilica-obtained carbonation of olivine in an autoclave under high pressure.

Production of metal carbonate and nanosilica below 100 nm was enabled in stirred bead milling, as reported by Wang and Forssberg [11]. It is shown that the stirred bead mill with very small beads can be used as efficient equipment for the production of the colloidal particles in the nanoscale from the feed materials of several microns in size at high energy consumptions. Generally, it is concluded that an intense comminution of carbonate minerals in the stirred bead mills leads to a progressive loss in crystallinity of the basal planes of the crystal structure. An intensive mechanical treatment of silica gives the structural changes and the amorphization.

Akhayere et al. [12] reported the synthesis of nanosilica from barley grass waste—an environmental burden—using varying temperatures during preparation. The temperatures used during the investigation were 400, 500, 600, and 700 °C, studying its effects on the mechanical properties of the nanosilica for use in environmentally friendly applications. Using the Brunauer–Emmett–Teller (BET) methodology, the surface area corresponds to 150 $m^2/g$. The results of this study showed improved and stable mechanical properties with the increase in temperature during synthesis.

A novel low-temperature vapor-phase hydrolysis method for the production of nanosilica using silicon tetrachloride was reported by Chen et al. [13]. Silica nanoparticles were obtained by the hydrolysis of silicon tetrachloride vapor with water vapor at a low temperature range (150–250 °C). Silica nanoparticles with a specific surface area of 418 $m^2/g$ and an average size of 141.7 nm were prepared at a temperature of 150 °C and with a residence time of 5 s. It was an amorphous mesoporous material, with an approximately spherical shape and a mass friction demission of 2.29.

Silica nanoparticles were prepared by ultrasonic spray pyrolysis (USP) between 300 and 600 °C using tetraethylorthosilicate (TEOS) as a precursor, as mentioned by Ratanathavorn et al. [14]. The particle size decreased from 347 to 106 nm when the synthesis temperature increased from 300 °C to 500 °C. Comparing two types of cream perfume, with and without silica, by applying cream perfume on a glass slide at 37 °C for 5 h, it was found that the odor of cream perfume with silica lasted longer than cream perfume without silica. The particles agglomerated and had irregular forms.

Citakovic [15] mentioned that the physical properties of nanomaterials have high significance for their application. Especially, a significant difference in the values of some physical parameters include the melting point, change in the unit-cell parameters, change

in the magnetic and optical characteristics, conductivity of the material, etc. The surface-to-volume ratio is an important parameter that has an impact on new characteristics in comparison to those of bulk materials. Silica is present in many different crystalline forms that vary in levels of fibrogenicity according to the degree of crystallization [16].

At present, nanosilica materials are prepared using several methods, including precipitation, sol–gel, acidic treatment, alkaline extraction, flow synthesis, stirred bead milling, the thermal decomposition technique, high-pressure carbonation, and low-temperature atmospheric pressure. However, with their high cost of preparation, many operations and morphological characteristics of particles have limited their wide application. By contrast, ultrasonic spray pyrolysis as a very simple method offers many advantages for the synthesis of oxidic particles as mentioned by Stopic et al. [17,18].

Generally, our aim was to reach the synthesis of nanosilica using the ultrasonic spray pyrolysis method. In contrast to previously mentioned work under high-pressure conditions in an autoclave [14], our aim was to obtain ideally spherical silica particles in a short residence time in dynamic conditions. In order to reach these aims, the concentration of precursor solution was adjusted for this purpose. Generally, an understanding of silica formation was studied via different calculations of the residence time and particle size. Regarding the previous literature analysis, our main aim is the testing of ultrasonic spray pyrolysis as a simple method for the synthesis of spherical silica particles suitable for lead treatment as mentioned by Huan [4].

## 2. Experimental Part

### 2.1. Material

The LUDOX 30 wt.% colloidal silica ($SiO_2$), SAFA 420811, VWR International GmbH, Darmstadt, Germany was used for preparation of precursor solution. The chosen volume of this concentrated solution was diluted in 900 mL of distillated water in order to prepare a suitable precursor for the synthesis of silica particles. The chemical analysis of solution was performed using ICP-OES analysis (SPECTRO ARCOS, SPECTRO Analytical Instruments GmbH, Kleve, Germany). The prepared precursor solution of different concentrations and the obtained suspension after the ultrasonic spray pyrolysis method was prepared for SEM and EDS analysis. First, the sample was shaken, a few drops were injected onto the aluminum slide with a pipette, allowed to dry, and finally evaporated with carbon together. A typical picture of this precursor material is shown in Figure 2:

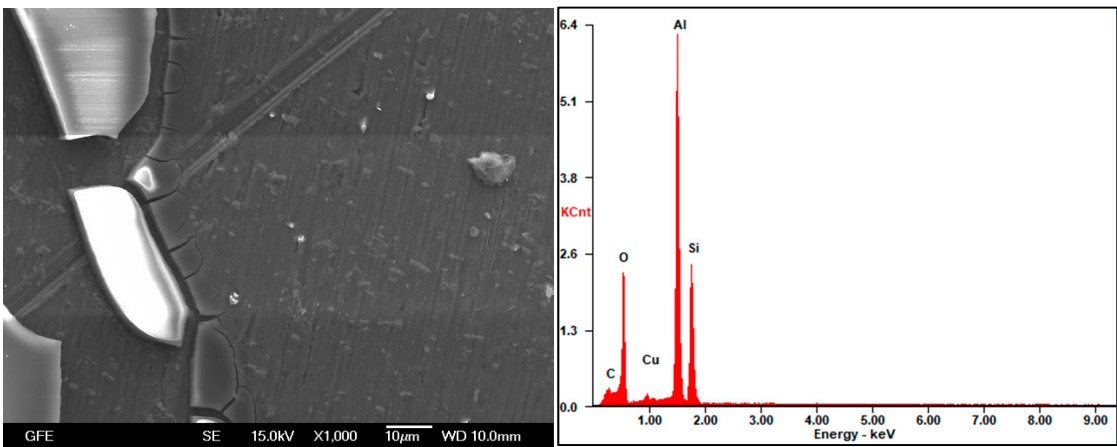

**Figure 2.** SEM and EDS analysis of diluted colloidal solution after evaporation.

The SEM analysis was performed on the JSM 7000F by JEOL (construction year 2006, JEOL Ltd., Tokyo, Japan) and EDX analysis using the Octane Plus-A by Ametek-EDAX (construction year, 2015, AMETEK Inc., Berwyn, PA, USA), with software Genesis V 6.53 by Ametek-EDAX, revealing an irregular structure of silica precursor, as shown in Figure 2. XRD analysis of silica powders was performed using a Bruker D8 Advance with a LynxEye

detector (Bruker AXS, Karlsruhe, Germany). X-ray powder diffraction patterns were collected on a Bruker-AXS D4 Endeavor diffractometer in Bragg–Brentano geometry, equipped with a copper tube and a primary nickel filter providing Cu K$\alpha$1,2 radiation ($\lambda$ = 1.54187 Å).

### 2.2. Procedure

Synthesis of silica was performed by the transformation of water solution of the chosen precursor to an aerosol in a strong ultrasonic field with an additional thermal decomposition of droplets at elevated temperatures in an inert atmosphere, as shown in Figure 3. The formed droplets of aerosol were transported with carrier gas to the laboratory tubular furnace (Ströhlein, Selm, Germany) in order to be transformed into nanosized particles. Due to the thermal stability of a quartz tube in a furnace, the maximal reaction temperature amounts to 1000 °C. The heating rate was 30 °C/min. Thermal decomposition of the precursor was performed at 900 °C. According to our previous work [17,18], an increase in gas temperature and aerosol velocity decreases the residence time of droplets in the reactor. A decrease in droplet size and an increase in gas temperature lead to a decreased particle size. The collection of powder was performed in two bottles filled with alcohol or distillated water. The obtained suspension was sent to SEM and EDS analysis. The scanning electron microscope was used to examine morphological characteristics of powders such particle sizes and shape. EDS analysis was performed for elemental analysis of the obtained silica powders.

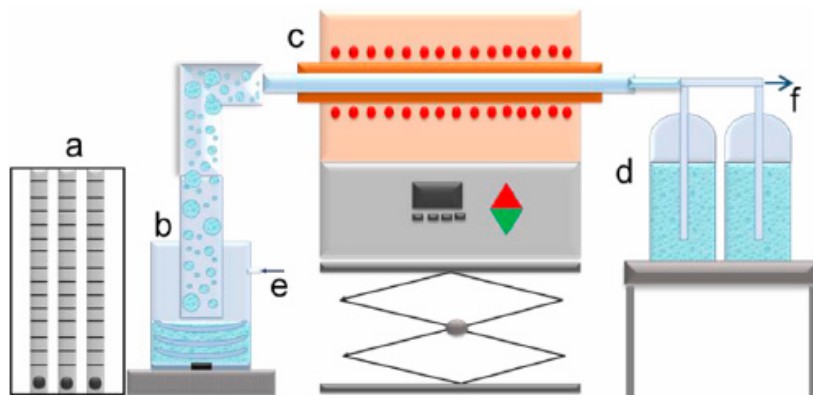

**Figure 3.** One-step ultrasonic spray pyrolysis lab-scale horizontal equipment: (**a**) Gas flow regulation; (**b**) ultrasonic aerosol generator; (**c**) furnace with the wall-heated reactor; (**d**) collection bottles; (**e**) gas inlet, (**f**) gas outlet.

Very fine aerosol droplets of precursor solution based on colloidal silica were obtained with an ultrasonic atomizer (PRIZNano, Kragujevac, Serbia), using three transducers with a frequency of 1.75 MHz in an ultrasonic field. The aerosol was carried with a nitrogen flow rate between 0.5 and 1.5 L/min into a quartz tube (1.0 m length and 0.021 m diameter) at 900 °C and placed in a previously mentioned Ströhlein furnace. The flow rate was measured using a special flowmeter gas unit (YOKOGAWA Deutschland GmbH, Ratingen).

### 2.3. Prediction of Particle Size

The formation of SiO$_2$ will be first defined via the diameter of an aerosol droplet ($d_d$), as shown with Equation (1) [19,20]:

$$d_d = 0.34 \left( \frac{8 \, \pi \sigma}{\rho_L f^2} \right)^{\frac{1}{3}}. \tag{1}$$

where: $d_d$—diameter of aerosol droplet, $f$—ultrasound frequency; $\rho_L$—density of water solution; $\sigma$—surface tension of water solution.

Using the following values: $f$—1.75 MHz; $\rho_L$—1.02 g/cm$^3$; $\sigma$—0.07 J/m$^2$, the calculated aerosol droplet amounts to 2.86 µm, as shown in Figure 4 [21]. As shown with

Equation (1) and Figure 4, the aerosol droplet can be decreased by increasing the ultrasonic frequency of an ultrasonic transducer.

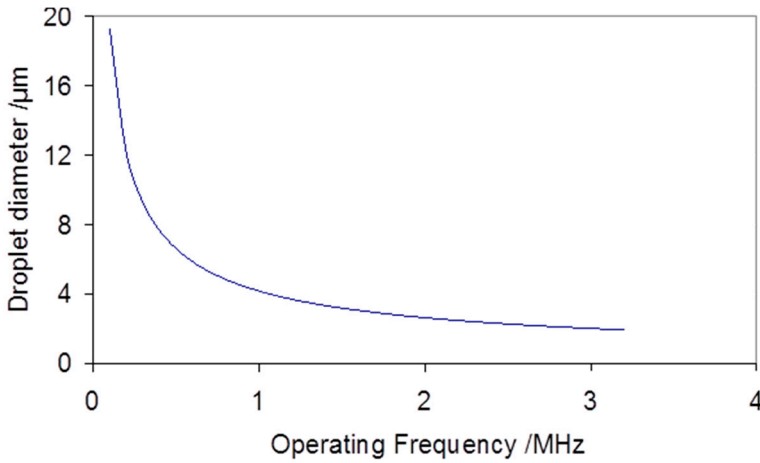

**Figure 4.** Dependence of aerosol droplet size of the operating frequency.

Assuming that the velocities of the droplet and carrier gas are equal, the droplet velocity was calculated from the ratio of the carrier gas flow ($q$) to the reaction zone area ($A$), as shown via Equation (2) [21].

$$v = \left( \frac{q}{A} \right) \tag{2}$$

Using $q$—0.5 dm$^3$/min and $A$—$0.38 \times 10^{-3}$ m$^2$, the droplet velocity amounts to 0.021 m/s.

According to our previous work, the residence time of droplets in the reaction tube can be calculated using Equation (3) [21].

$$t = \left( \frac{V \cdot \text{To}}{q \cdot \text{Tr}} \right) \tag{3}$$

where: $V$—volume of heating zone in tube (m$^3$); $q$—flow rate (L/min); Tr—reaction temperature (K); and To—room temperature. Using the following values: $v$—$0.21 \times 10^{-3}$ m$^3$, $q$—0.5–1.5 L/min; Tr—1173 K; and To—298 K, the residence times were calculated at room temperature and 1173 K, as shown in Figure 5. This residence time depends on the reaction temperature and flow rate of carrier gas. An increase in flow rate from 0.5 to 2.5 L/min leads to a residence time of a few seconds in the tubular reactor. An increase in temperature from 25 °C to 900 °C decreases the residence time from 14 to 4 s using a flowrate of 0.5 L/min.

The particle size ($d_p$) depends on the droplet size and concentration of solution ($c$). This correlation between the concentration and other precursor characteristics and the final particle size, under the assumption that no precursor is lost in the process, can be described with the following Equation (4) derived from one basic equation reported by Messing et al. [22]:

$$d_p = d_d \left( \frac{M_p}{M_{SiO_2}} * \frac{C}{\rho} \right)^{0.33} \tag{4}$$

where $d_p$ is the diameter of the particle, $d_d$ is the diameter of the aerosol droplet, $M_p$ is the molar mass of the precursor (g/mol), $\rho$ is the density of silica particles, and $c$ is the concentration of the precursor solution.

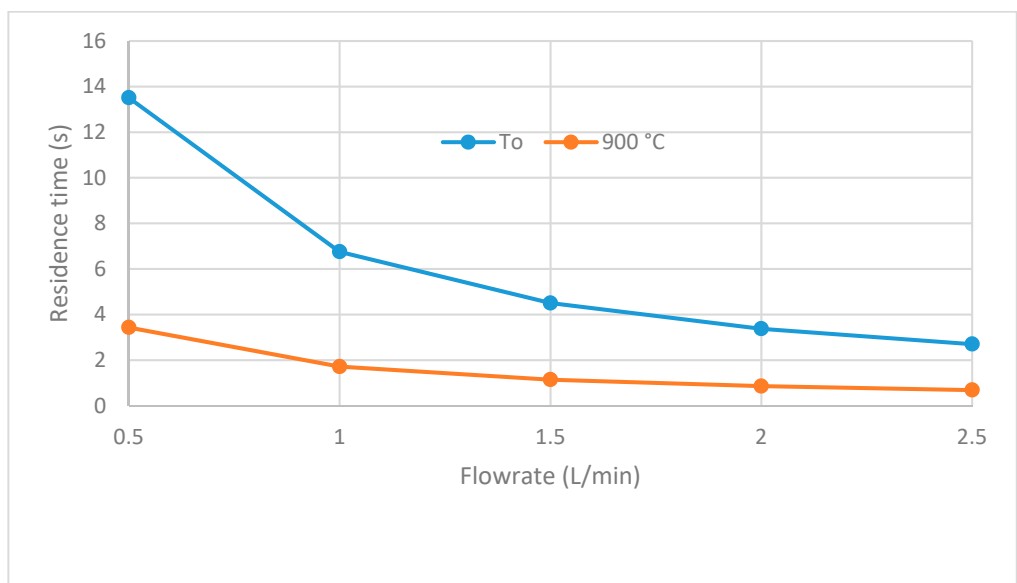

**Figure 5.** Calculated values of the residence time at different temperatures.

Using the following values: $M_{SiO2}$ = 60.09 g/mol; $\rho_{SiO2}$ = 2.65 $\times$ 10³ kg/m³; and concentrations of solution (g/cm³): 60, 30, 15, 7.5, 1.5, 0.15, and 0.1, the obtained values for particles sizes are presented in Table 1:

**Table 1.** Calculated particle size depending on concentration of precursor solution.

| Concentration (mol/L) | 1 | 0.5 | 0.25 | 0.125 | 0.025 | 0.0025 | 0.0017 |
|---|---|---|---|---|---|---|---|
| Concentration (g/cm³) | 60 | 30 | 15 | 7.5 | 1.5 | 0.15 | 0.10 |
| Particle size (nm) | 810 | 643 | 508 | 412 | 283 | 110 | 96 |

The obtained values of particle sizes are presented in Figure 6.

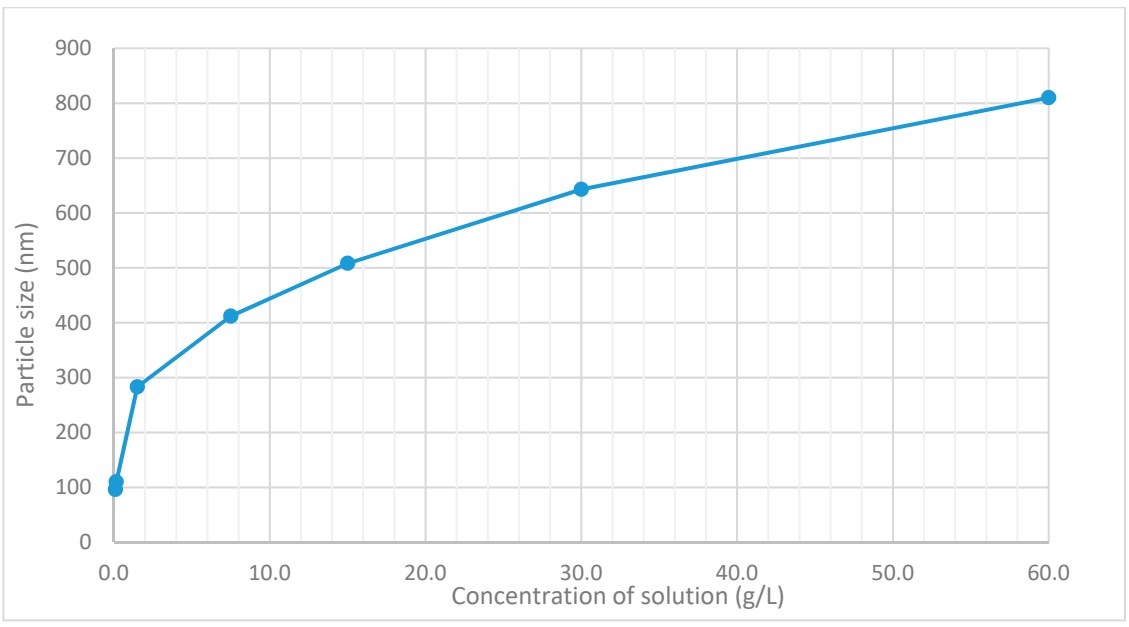

**Figure 6.** Relationship between particle size and concentration of precursor solution.

As expected by Equation (4), a decrease in the precursor concentration of solution decreases the particle size. A comparison of the calculated data with those published in the literature by Kim [7] confirmed that ultrasonic spray pyrolysis can also prepare particle sizes of 100 nm using a small concentration of solution of 0.1 mg/L (0.002 mol/L). In order to validate the prediction of particle size from Table 1, the following experimental concentrations (0.5 and 0.125 mol/L) were tested in our experimental work.

## 3. Results and Discussion

SEM and EDS analysis of the obtained particles at 900 °C using precursor solution concentrations of 0.50 and 0.125 mol/L found very fine spherical particles after ultrasonic spray pyrolysis, as shown in Figures 7a and 8a. Using small concentrations of solution such as 0.125 mol/L, the silica particle is ideally spherical, and single without agglomeration, which is a typical case for higher concentrations. The presence of large particles of about 1 μm is confirmation of a collision of droplets during the transport of an aerosol. Some satellite spherical particles are observed at primary large particles. The measurement of particle size was performed using Software Image Pro Plus, Media Cybernetics, USA [23].

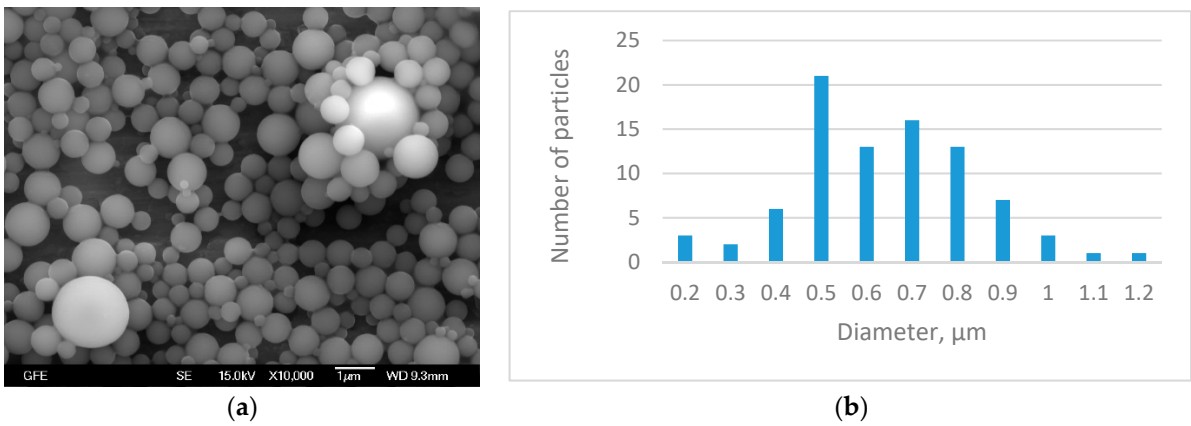

(**a**)  (**b**)

**Figure 7.** (**a**) SEM analysis of particles obtained at 900 °C using 0.50 mol/L precursor solution. (**b**) Particle size distribution.

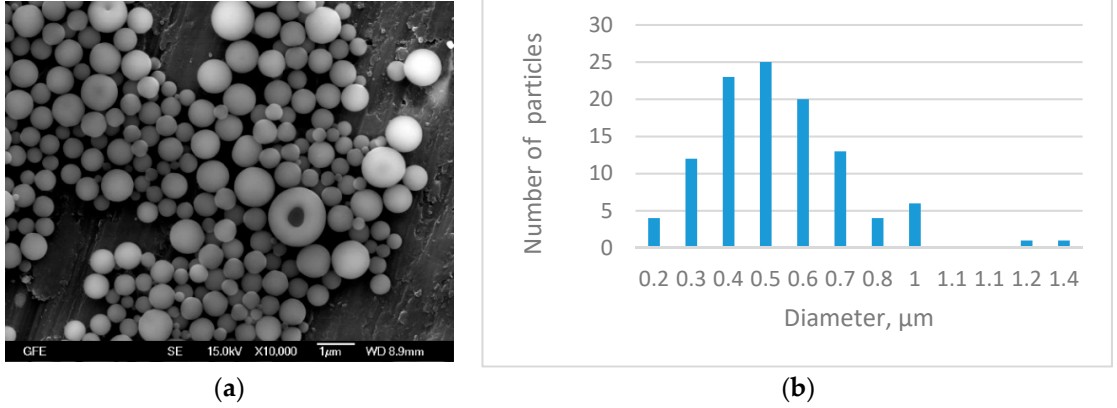

(**a**)  (**b**)

**Figure 8.** (**a**) SEM analysis of particles obtained at 900 °C using 0.125 mol/L precursor solution. (**b**) Particle size distribution.

A decrease in solution concentration from 0.5 to 0.125 mol/L leads to smaller particle size, as shown in Figures 7b and 8b, respectively. EDS analysis has confirmed the presence of silicon and oxygen together with elements such as Al, Cu, and C, which are used for the preparation of samples for characterization, as shown in Figure 9.

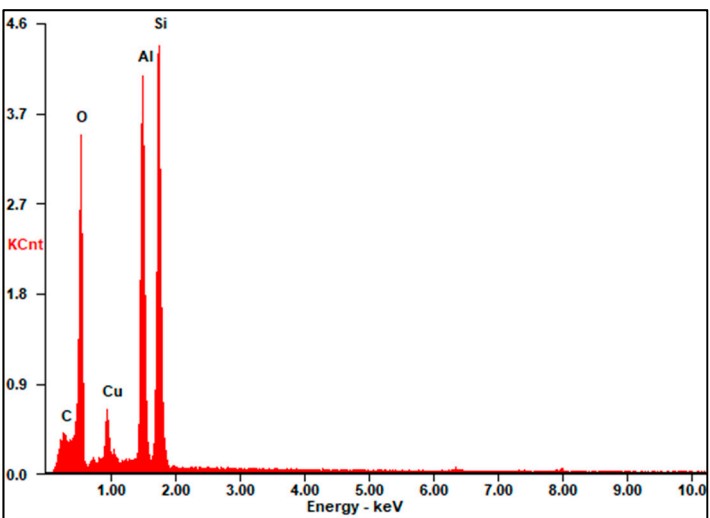

**Figure 9.** Typical EDS analysis for both obtained powders.

The presence of elements such as Cu, C, and Al are not connected with our ultrasonic spray pyrolysis synthesis. Additionally, ICP-OES analysis of Si in solution before and after USP synthesis was included in our consideration. The concentration of Si decreased from 13,600 to 74.6 mg/L for the 0.5 M precursor solution. A similar behavior was revealed for the 0.125 M solution, where the concentration of Si decreased from 3750 to 1 mg/L, which confirms the full transformation of the used precursor to $SiO_2$.

For the sample with a concentration of 0.5 mol/L, most particles are around 0.6 μm in diameter, with an average diameter value of 0.68 μm, which suggests similar particle diameters. The sample with a concentration of 0.125 mol/L could be characterized with most particles around 0.50 μm and with an average diameter 0.60 μm, which suggests the existence of larger particles but with a lower quantity, and the lower diameter was compared with particles from the precursor solution of 0.50 mol/L. The calculated particle sizes are situated between the maximal and minimal values of the measured diameters, as shown in Table 2.

**Table 2.** The values of theoretical and measured silica particles.

| Concentration of Solution, mol/L | Calculated Value of Diameter, μm | Measured Maximal Diameter, μm | Measured Average Diameter, μm | Measured Minimal Diameter, μm |
|:---:|:---:|:---:|:---:|:---:|
| 0.50 | 0.64 | 1.20 | 0.69 | 0.16 |
| 0.125 | 0.41 | 1.35 | 0.61 | 0.24 |

According to our previous laser diffraction measurement of produced aerosol from an ultrasonic generator between 0.8 and 2.5 MHz reported by Bogovic et al. [24], the obtained values of droplet size are, in all cases, higher than theoretically predicted, as shown with Equation (1), due to the immediate coagulation that occurs in the aerosol production chamber. As mentioned previously by Tsi et al. [25], only 5–10% of the particle sizes obtained in spray pyrolysis of 6–9 μm precursor droplets were of the sizes predicted by the one-particle-per-droplet mechanism. Differences between calculated and experimentally obtained values of particle sizes may be partially due to the approximate values used for surface tension and the density of aqueous solution, micro-porosity of particles, and mostly due to the coalescence/agglomeration of aerosol droplets at a high flow rate for the carrier gas (turbulence effects). In Equation (4) [22,26], also based on the assumption of one particle per droplet, the influence of temperature on the mean particle size was not taken into account.

XRD analysis of powder obtained at 900 °C has shown an amorphous structure of the prepared silica powder, as shown in Figure 10. A Hill-like peak in the range of [2Θ] = 21–24 indicates the absence of any ordered crystalline structure and a highly disordered structure of silica. The same X-ray diffraction patterns of nanosilica were reported by Huan et al. [4]. The extracted nanosilica from tetraethoxysilane (TEOS) has a high lead treatment efficiency from waste water. We hope that spherical amorphous silica particles prepared by ultrasonic spray pyrolysis have the same properties. Chen et al. [13] reported from a comparison with crystalline silica that the amorphous structure has more advantages such as nontoxicity, better interaction, and pollution adsorbent. In comparison to particles obtained by Huan [4], we produced ideally spherical nonagglomerated particles.

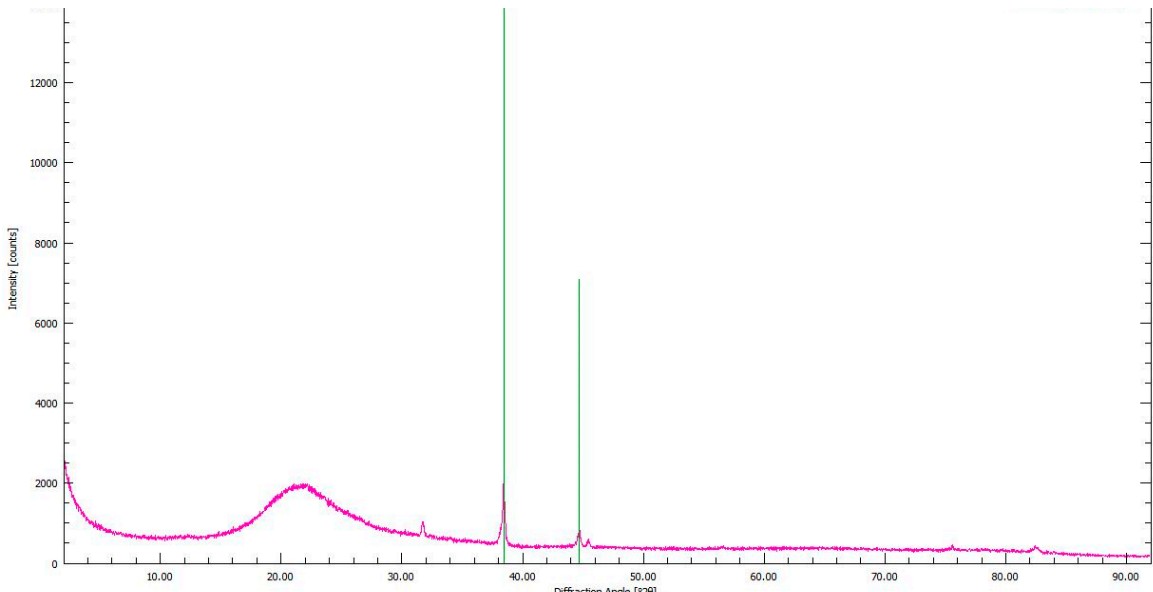

**Figure 10.** XRD analysis of silica powder obtained by ultrasonic spray pyrolysis.

The crystalline peaks in Figure 10 belong to aluminum (sample holder). In order to obtain a fully crystalline structure, ultrasonic spray pyrolysis shall be performed at higher temperatures and longer residence times. According to Figures 5 and 6, we found that the chosen residence time in the furnace is sufficient for the complete transformation of precursor to the aimed $SiO_2$. Regarding the synthesis of silica using the previously mentioned sol–gel, high-pressure carbonation, and other methods, ultrasonic spray pyrolysis enables a controlled particle size and morphology using different concentrations of solution.

## 4. Conclusions

Synthesis of silica powder was performed from a high concentrated colloidal solution (30%) at 900 °C using the ultrasonic spray pyrolysis method. This method enables the production of very fine spherical silica particles from an irregular structure in one horizontal reactor. The controlled synthesis of particles was reached by changing the concentration of precursor solution from 0.5 to 0.125 mol/L. A decrease in concentration from 0.5 to 0.125 mol/L leads to a decrease in measured average diameter from 690 to 610 nm. A Hill-like peak in the range of [2Θ] = 21–24 obtained by XRD analysis indicates the absence of any ordered crystalline structure and a highly disordered structure of silica with high purity. A comparison of theoretically and measured diameter values of prepared silica has shown relatively good agreement, where a deviation amounts to 17% for average diameter. A disadvantage of this method is the collision of droplets during their transport using carrier gas, and especially low efficiency, due to losses in the dissolved precursor on the construction elements of the reactor.

**Author Contributions:** Conceptualization, F.W. and S.S.; funding acquisition, B.F.; investigation, F.W.; methodology, S.S. and F.W.; supervision, S.S. and B.F.; writing—original draft, F.W, T.-V.H. and S.S. All authors have read and agreed to the published version of the manuscript.

**Funding:** This research received no external funding.

**Institutional Review Board Statement:** Not applicable.

**Informed Consent Statement:** Not applicable.

**Conflicts of Interest:** The authors declare no conflict of interest.

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
