# Peer review of "Synthesis of Silica Particles Using Ultrasonic Spray Pyrolysis Method"

_metals, doi:10.3390/met11030463_

Round 1

Reviewer 1 Report

Paper entitled “ Synthesis of silica particles using ultrasonic spray pyrolysis method” meets the necessary standards for publication in this journal.

I recommend: 

-        Attention when writing references. For example, at point 2.1. line 3, the word "ptrcursor" is misspelled.  also "ml" is misspelled.-        Perhaps it would have been unnecessary to make a comparison with other similar materials or synthesis methods. What are the advantages and disadvantages of using other materials or other synthesis methods in terms of particle size?

Please check the entire manuscript carefully for eventual typographical errors.
Final Conclusion: The paper meets the necessary standards for publication.

Author Response

Dear Reviewer, thank you very much for your invested time and valuable comments. We improved our text using the comments from 3 Reviewers.

Attention when writing references.

We added 10 new References and improved the present references

For example, at point 2.1. line 3, the word "ptrcursor" is misspelled.  also "ml" is misspelled.-       

We have written "precursor" and "mL".

Perhaps it would have been unnecessary to make a comparison with other similar materials or synthesis methods.

At present, nanosilica materials are prepared using several methods, including precipitation, sol-gel, acidic treatment, alkaline extraction, flow synthesis, stirred bead milling, thermal decomposition technique, high pressure carbonation, and low temperature atmospheric pressure method. However, their high cost of preparation, many operations and morphological characteristics of particles have limited their wide application. In contrast, ultrasonic spray pyrolysis as very simple method offering many advantages for synthesis of oxidic particles as mentioned by Stopic et al [17, 18]. Therefore, our aim was to reach a synthesis of nanosilica using ultrasonic spray pyrolysis method. 

What are the advantages and disadvantages of using other materials or other synthesis methods in terms of particle size?

In comparison to  ultrasonic spray pyrolysis an Advantage of an using of other methods such as sol-gel, thermal decomposition, acidic Treatment and stirred bead milling enables formation very small nanosized particles below 100 nm. Disadvantage of using other methods in multistep process is formation of the agglomerated irregular form in contrast to ideally spherical particles obtained by ultrasonic spray pyrolysis. A limited choice of precursor for the Synthesis of silica is a disadvantage of this method.

Please check the entire manuscript carefully for eventual typographical errors.

We checked and improved entire manuscript carefully for present typographical Errors.

I hope that you will be satisfied with new version.

Thank you for your support.

Reviewer 2 Report

The manuscript describes the synthesis of silica nanoparticles using ultrasonic spray coupled with pyrolysis method. This paper needs a lot of improvements before being accepted in Metals.

  1. The command of English is very poor. Many spelling and grammatical errors can be easily found in the entire manuscript. Hence, this paper has to be sent to professional English editing service for proofread.
  2. A decision has to be made on which of two variants of English to adopt (British or American) so that there is consistency in usage.
  3. Many technical errors are detected. For instance, 700oC should be 700 °C (check whole manuscript), 5 hours should be 5 h, subscripts and superscripts in chemical compounds, etc.
  4. Abbreviations should be defined before use.
  5. What is the problem statement that leads the authors to perform this work? It only described very brief in the Introduction.
  6. Page 4, lines 2-5: Remove those unrelated sentences.
  7. Section 2 Experimental part has to rewrite because some results are discussed in this section. Also many important information about methodology is missing in this section.
  8. Page 5, line 1 from bottom: reference for equation (1) should be inserted. Some symbols in equations are not defined.
  9. Figures 7 and 8: The particle size distributions have to replot (number of particles vs diameter (nm)). Current plots cannot give much information.
  10. Why the sample has Al, Cu and C elements? EDS is not a proper technique to quantize the chemical composition. Please put XRF or ICP-OES spectroscopy data.
  11. The discussion of results is very plain and lack of information. No references to support the results and observation. Overall the discussion is dull and not interesting. Hence, I suggest a thorough re-write in this section.
  12. What is the yield of silica nanoparticles?
  13. Table 2. Figures with 2-3 decimal points are sufficient.
  14. Figure 10: Why the amorphous sample has crystalline peaks?
  15. Page 10, line 5: Please describe more and support with experimental data.
  16. I am not convinced with theoretical and experimental values of diameter of silica particles. Only particle size synthesized using 2 concentrations are reported. More data are needed to make this work more complete and informative.

Author Response

Dear Reviewer,

thank you very much for your invested time and  very important comments in order to improve this paper. Our changes are written in red Color in our new Version!

  1. The command of English is very poor. Many spelling and grammatical errors can be easily found in the entire manuscript. Hence, this paper has to be sent to professional English editing service for proofread. We changed the gramamatical errors in our text. This paper will be sent necessarily to professional English editing service for proofread!
  2. A decision has to be made on which of two variants of English to adopt (British or American) so that there is consistency in usage. We have used British variant of English. Finally, It will be improved by English editing Service at our Institute!
  3. Many technical errors are detected. For instance, 700oC should be 700 °C (check whole manuscript), 5 hours should be 5 h, subscripts and superscripts in chemical compounds, etc. We found many similiar mistakes and  changed it in our text. Please to see words in red Color in our new text.
  4. Abbreviations should be defined before use. We defined abbrevations  before use (SEM, EDS, XRD, USP) scanning electron microscope (SEM) and energy dispersive X-ray spectroscopy (EDS). X-Ray Diffraction, frequently abbreviated as XRD, Ultrasonic spray Pyrolysis (USP)
  5. What is the problem statement that leads the authors to perform this work? It only described very brief in the Introduction. Our aim was a controlled Synthesis of very fine spherical silica particles using ultrasonic spray pyrolysis method, what is missing in literature. We tested firstly high pressure leaching in an autoclave, but we did not reach our aim.

    At present, nanosilica materials are prepared using several methods, including precipitation, sol-gel, acidic treatment, alkaline extraction, flow synthesis, stirred bead milling, thermal decomposition technique, high pressure carbonation, and low temperature atmospheric pressure method. However, their high cost of preparation, many operations and morphological characteristics of particles have limited their wide application. In contrast, ultrasonic spray pyrolysis as very simple method offering many advantages for synthesis of oxidic particles as mentioned by Stopic et al [17, 18]. Therefore our aim was to reach a synthesis of nanosilica using ultrasonic spray pyrolysis method. 

  6. Page 4, lines 2-5: Remove those unrelated sentences. We removed these sentences "Exposure to airborne nanoparticles contributes to many chronic pulmonary diseases. Silicosis is caused by the inhalation of crystalline silica particles for extended periods of time."
  7. Section 2 Experimental part has to rewrite because some results are discussed in this section. Also many important information about methodology is missing in this section. In Experimental part we discussed results about our precursor material.Although  SEM/EDS Analysis represents new  result, we found that we have firstly to describe our precursor. Therefore some results about material are discussed firstly in this section.
  8. Page 5, line 1 from bottom: reference for equation (1) should be inserted. Some symbols in equations are not defined. We inserted two references for Equation 1/Lang, R.J. Ultrasonic atomization of liquids, Journal of Acoustical Society of America 1962, 34, 7-10. and /Peskin, R.L, Raco, R.J. Ultrasonic atomization of liquids, Journal of Acoustical Society of America 1966, 35, 1378-1382.  We defined the diameter of aerosol droplet (dd) in Eq. 1.
  9. Figures 7 and 8: The particle size distributions have to replot (number of particles vs diameter (nm)). Current plots cannot give much information. We have reploted The particle size Distribution  at Figures 7 and 8 and included new diagrams in our text. You had right. It Looks now better.
  10. Why the sample has Al, Cu and C elements? EDS is not a proper technique to quantize the chemical composition. Please put XRF or ICP-OES spectroscopy data. Because of using of different carriers from Al, Cu and C, our EDS Analysis has confirmed the presence of Al, Cu and C. These elements are not connected with our ultrasonic spray pyrolysis Synthesis. As proposed from your side Additionally we have included ICP-OES Analysis of solution before and after Synthesis. "ICP_OES Analysis of  Si in precursor solution was performed before and after the ultrasonic spray pyrolysis synthesis. The concentration of Si was decreased from 13600 mg/L to 74.6 mg/l for 0.5 M precursor solution. Similar behavior was obtained for 0.125 M solution, where the concentration of Si was decreased from  3750 mg/L to 1 mg/L, what confirms fully transformation of the used Precursor to SiO2.
  11. The discussion of results is very plain and lack of information. No references to support the results and observation. Overall the discussion is dull and not interesting. Hence, I suggest a thorough re-write in this section. In discusion of our results we added new 10 references and additional comments, in order to Support our results and observations. Lang, R.J. Ultrasonic atomization of liquids, Journal of Acoustical Society of America 1962, 34, 7-10. Peskin, R.L, Raco, R.J. Ultrasonic atomization of liquids, Journal of Acoustical Society of America 1966, 35, 1378-1382. /Bogovic, J, Schwinger, A., Stopic, S., Schroeder, J., Gaukel, V., Schuhmann, P., Friedrich, B. (2011): Controlled droplet size distribution in ultrasonic spray pyrolysis, Metall 2011, 10, 455-459./Tsai, S.C., Song, Y.L., Tsai, C.S., Yang, C., Chiu. W., Lin. H., Ultrasonic spray pyrolysis for nanoparticles synthesis. Journal of Materials Science 2004, 39, 3647–3657/Messing, G., Zhang, S., Jayanthi, G. Ceramic powder synthesis by spray pyrolysis, Journal American Ceramic Society 1993, 76, 2707-2726/Stopić, S, Friedrich, B, Dvorak, P. Synthesis of nanosized spherical silver powder by ultrasonic spray pyrolysis, Metall 2006, 60, 6, 377-382. Also we added new comments in introduction and about our results.
  12. What is the yield of silica nanoparticles? Firstly, ICP-OES Analysis of Silicon of Solution has indirectly confirmed the fully transformation of precursor to SIO2. XRD has directly confirmed the yield of 100 % of silica nanoparticles. XRD analysis of powder obtained at 900 °C, has shown an amorphous structure of the prepared silica powder, as shown at Figure 10. Hill like peak in the range of [2Ө]= 21-24, indicates the absence of any ordered crystalline structure and highly disordered structure of silica. Identical X-ray diffraction pattern of nanosilica high purity were reported by Huan et al. [4]. 
  13. Table 2. Figures with 2-3 decimal points are sufficient.You have right. We have changed it in Table 2 and put it in our text.
  14. Figure 10: Why the amorphous sample has crystalline peaks? The crystalline peaks belong to Aluminium (sample holder).
  15. Page 10, line 5: Please describe more and support with experimental data. According to our previous laser diffraction measurement of produced aerosol [21] from an ultrasonic generator between 0.8 and 2.5 MHz, obtained values of droplet size are in all cases higher than theoretically predicted as shown with Eq. 1, due to immediate coagulation that occurs in the aerosol production chamber. Ref.21: Bogovic, J, Schwinger, A., Stopic, S., Schroeder, J., Gaukel, V., Schuhmann, P., Friedrich, B. (2011): Controlled droplet size distribution in ultrasonic spray pyrolysis, Metall, 10, 455-459.
  16. I am not convinced with theoretical and experimental values of diameter of silica particles. Only particle size synthesized using 2 concentrations are reported. More data are needed to make this work more complete and informative. You have right. More data  than 2 used concentrations are needed to make this work more complete and informative. We found that our present results can be interesting for our colleagues.We based our work on our 20 years experience on the ultrasonic Synthesis spray pyrolysis, as published in our first Special issue: "Advances in Synthesis of Metallic, oxidic and Composite powders-https://www.mdpi.com/books/pdfview/book/3390 We have just published  some influences of parameters (temperature, flow rate, and concentration of solution) on particle sizes in our previous publications [17, 18]. Therefore, we  did not perform a lot of experiments.

I hope that you will be satisfied with new improved Version.

Thank you for your support!

Reviewer 3 Report

Review of manuscript

“Synthesis of silica particles using ultrasonic spray pyrolysis method”

The paper topic is actual, and authors obtained interesting data which need to be more deeply analysed and carefully presented for publication

Some comments are below:

Abstract:

Authors mentioned: Synthesis of silica particles was performed at 900°C using ultrasonic spray pyrolysis method. It seems to be reasonable to show in abstract some advantages of this method as compared to those being used in industry.

  1. Introduction
    • There is no comparison of Mohanray at al. [3]route with another methods described
    • There is no analysis of drawbacks and advantages of Huan et al. [4] sol-gel method and Kim et al. [7] technology which use acid treatment and surface modification from blast-furnace slag
    • Page 3, line 2: to correct English of sentence: “…This process is an environment friendly processes apsorbing carbon dioxide”
    • Page 3, line 4-33: Authors chaotically describe the synthesis methods without their comparison and conclusions. It is impossible to understand why authors decided to choose ultrasonic method
    • Page 4, lines 6-8: The paper aim “….to reach a synthesis of nanosilica using ultrasonic spray pyrolysis method” is not clear because of lacks of previous description of the methods. The similar impression is for statement “…. In contrast to previously mentioned work under high pressure conditions in an autoclave [14], our aim was to obtain ideally spherical particles silica in short residence time in dynamic conditions”. Authors need to specify in detail this goal based on previous analysis of the synthesis methods
  1. Experimental part
    • Page 4, line 15: English error in sentence”… The chosen volume of this concentrated solution was diluted in 900 ml distillated water in order to prepare suitable ptrcursor for the synthesis of silica submicron particles.”
    • Page 5, Line 18: To correct English: “ The prepared precursor solution of different concentration and obtained suspension after suspencion after ultrasonic spray pyrolysis method was prepared for SEM and EDS- Analysis.
    • Page 5, lines 3-5: The sentence “…Synthesis of silica was performed by one step ultrasonic spray pyrolysis transforming of water solution of chosen precursors to an aerosol in a strong ultrasonic field in an ultrasonic atomizer as shown at Figure 3” is difficult to understand. Pyrolisisis the thermal decomposition of materials at elevated temperatures in an inert atmosphere. It involves a change of chemical composition. So, “ultrasonic spray pyrolysis” means application of ultrasonic field during heating. Authors talk about ultrasonic atomizing of  “water solution of chosen precursors”, not  “ultrasonic pyrolysis”. Please, clarify.
    • Page 5, lines 5-8: The description of Fig.3 “…The formed droplets of aerosol were transported with carrier gas to the laboratory tubular furnace (Strohlein, Selm, Germany) in order to be transformed in nanosized particles. The residence time of aerosol in an furnace and concentration of solution with reaction temperature have an important influence on the morphology and size of aimed product.” is not understandable. Authors need to correct English and add more detailed description of the reactor and its working parameters. For example, authors need to answer question “ How do the droplet diameter and other parameters (gas temperature, aerosol velocity, etc) influence on the particle size?
    • Page 5, line 28. It is unclear why authors talk about SiC: “…The formation of SiC will be firstly defined via the diameter of aerosol droplet (dd) as shown with equation (1)’’. Is Eq1 is authors or [17] equation? Please, specify.
    • Page 6, lines 3-20: Authors use the formulas without any explanations and citing. Moreover, the choice of various parameters value is not proven and described.
    • Page 7-8, Table1, Fig.6 : There is no comparison of calculated data with those published in the literature
    • Page 9, line 2: Authors state: ‘’ …A decrease of solution concentration leads to smaller particle size as shown at figure 8.’’ However, particle size dependence on solution concentration is not shown on Fig.8
    • Page 9, lines 3-6: Please, explain the sentence “…EDS analysis has confirmed the presence of silicon and oxygen togehter with elements such as Al, Cu, C, which are used for the preparation of samples for characterisation, as shown at Figure 9.”
    • Page 9, line 20: Looks like the term “Theoretical particle size“ means calculated particle size in the sentence “…..Theoretical particle size are situated in frame of maximal and minimal values of measured diameters, as shown in Table 2’. Please correct and edit English

  1. Conclusion needs to be corrected and detailed to present some real data obtained by authors

Author Response

Dear Reviewer,

thank you very much for your invested time and many valuable comments in order to improve our paper. In my new Version I included your commens and written my answers in red Color.

Authors mentioned: Synthesis of silica particles was performed at 900°C using ultrasonic spray pyrolysis method. It seems to be reasonable to show in abstract some advantages of this method as compared to those being used in industry.

In comparison to other methods such as sol-gel, acidic treatment, thermal decomposition, stirred bead milling, and high pressure carbonation, the advantage of the ultrasonic spray method for preparation of nanosized silica controlled morphology is the simplicity of setting up individual process segments and changing their configuration, one step continuous synthesis, and the possibility of synthesising nanoparticles from various precursor.

  1. Introduction
    • There is no comparison of Mohanray at al. [3] route with another methods described. We made it in our new Version:
    • Unfortunatelly, starting from corn cob ash Calcination and mixing with NaOH [3],  precipitation with  1 % polyvinil alcohol (PVA) can not ensure the controlled silica particle size and their purity. The impurities originate from calcination process. 
    • There is no analysis of drawbacks and advantages of Huan et al. [4] sol-gel method and Kim et al. [7] technology which use acid treatment and surface modification from blast-furnace slag.
    • We made it: XRD analysis of powder obtained at 900 °C, has shown an amorphous structure of the prepared silica powder, as shown at Figure 10. Hill like peak in the range of [2Ө]= 21-24, indicates the absence of any ordered crystalline structure and highly disordered structure of silica. Same X-ray diffraction pattern of nanosilica were reported by Huan et al. [4]. The extracted nanosilica from tetraethoxysilane (TEOS) has high lead treatment efficiency from waste water. 
    •  An acidic Treatment as shown in [7] can lead to Formation of silica gel and blockage of whole process. The filtration is a required operation in this process. This Treatment contains many operations in order to obtain silica. Because of these characteristics we need fo find other simple method for the synthesis of very fine silica without an acidic treatment.
    • Page 3, line 2: to correct English of sentence: “…This process is an environment friendly processes apsorbing carbon dioxide” This method is an environment friendly process related to the capture od carbon dioxide and preparation of silica.
    • Page 3, line 4-33: Authors chaotically describe the synthesis methods without their comparison and conclusions. It is impossible to understand why authors decided to choose ultrasonic method, We gave additional informations and Advantage of an using of USP-Method.
    • At present, nanosilica materials are prepared using several methods, including precipitation, sol-gel, acidic treatment, alkaline extraction, flow synthesis, stirred bead milling, thermal decomposition technique, high pressure carbonation, and low temperature atmospheric pressure However, their high cost of preparation, many operations and morphological characteristics of particles have limited their wide application. In contrast, ultrasonic spray pyrolysis as very simple method offering many advantages for synthesis of oxidic particles as mentioned by Stopic et al [17, 18].  
    •  
    • Page 4, lines 6-8: The paper aim “….to reach a synthesis of nanosilica using ultrasonic spray pyrolysis method” is not clear because of lacks of previous description of the methods. The similar impression is for statement “…. In contrast to previously mentioned work under high pressure conditions in an autoclave [14], our aim was to obtain ideally spherical particles silica in short residence time in dynamic conditions”. Authors need to specify in detail this goal based on previous analysis of the synthesis methods
    • Regarding to the previous analysis our main aim is testing of ultrasonic spray pyrolysis as simple method for Synthesis of spherical silica particles suitable for lead Treatment as mentioned by Huan [4].
  1. Experimental part
    • Page 4, line 15: English error in sentence”… The chosen volume of this concentrated solution was diluted in 900 ml distillated water in order to prepare suitable ptrcursor for the synthesis of silica submicron particles.”
    • we changed ptrcursor in precursor
    • Page 5, Line 18: To correct English: “ The prepared precursor solution of different concentration and obtained suspension after suspencion after ultrasonic spray pyrolysis method was prepared for SEM and EDS- Analysis.
    • I changed it:
    • The prepared precursor solution of different concentration and obtained suspension  after ultrasonic spray pyrolysis method was prepared for SEM and EDS- Analysis.
    • Page 5, lines 3-5: The sentence “…Synthesis of silica was performed by one step ultrasonic spray pyrolysis transforming of water solution of chosen precursors to an aerosol in a strong ultrasonic field in  an ultrasonic atomizer as shown at Figure 3” is difficult to understand. Pyrolisisis the thermal decomposition of materials at elevated temperatures in an inert atmosphere. It involves a change of chemical composition. So, “ultrasonic spray pyrolysis” means application of ultrasonic field during heating. Authors talk about ultrasonic atomizing of  “water solution of chosen precursors”, not  “ultrasonic pyrolysis”. Please, clarify.. We changed it:
    • Synthesis of silica was performed by transforming of water solution of chosen precursors to an aerosol in a strong ultrasonic field with an additional thermal decomposition of droplet at elevated temperatures in an inert atmosphere
    •  
    • Page 5, lines 5-8: The description of Fig.3 “…The formed droplets of aerosol were transported with carrier gas to the laboratory tubular furnace (Strohlein, Selm, Germany) in order to be transformed in nanosized particles. The residence time of aerosol in an furnace and concentration of solution with reaction temperature have an important influence on the morphology and size of aimed product.” is not understandable. Authors need to correct English and add more detailed description of the reactor and its working parameters. For example, authors need to answer question “ How do the droplet diameter and other parameters (gas temperature, aerosol velocity, etc) influence on the particle size? We changed it.
    •  The formed droplets of aerosol were transported with carrier gas to the laboratory tubular furnace (Ströhlein, Selm, Germany). Because of  a thermal stability of a quartz tube in a furnace, the maximal reaction temperature amounts 1000°C. The heating rate was 30°C/min. Thermal decomposition of precursor was performed at 900°C. According to our previous work [17, 18], an increase of gas temperature and Aerosol Velocity decreases the residence time of droplet in reactor. A decrease of droplet size and an increase of  gas temperature lead to a decreased particle size.
    •  
    • Page 5, line 28. It is unclear why authors talk about SiC: “…The formation of SiC will be firstly defined via the diameter of aerosol droplet (dd) as shown with equation (1)’’. Is Eq1 is authors or [17] equation? Please, specify.
    • SiC is mistake. SiO2 is correct. We changed it.
    • For Eq. 1 we added references [19, 20] 
    • Lang, R.J. Ultrasonic atomization of liquids, Journal of Acoustical Society of America 1962, 34, 7-10.
    • Peskin, R.L, Raco, R.J. Ultrasonic atomization of liquids, Journal of Acoustical Society of America 1966, 35, 1378-1382.
    •  
    • Page 6, lines 3-20: Authors use the formulas without any explanations and citing. Moreover, the choice of various parameters value is not proven and described.
    • We have used equations 3 and 4 and the choice of various parameters according to Publication [21]. We included a new citation.
    • 21. Stopic, S: Synthesis of metallic nanosized particles by ultrasonic spray pyrolysis, Publisher: Shaker GmbH, Kohlsheid, 2015, 117.
    • For Eq. 5 we added a new reference:
    • 22. Messing, G., Zhang, S., Jayanthi, G. Ceramic powder synthesis by spray pyrolysis, Journal American Ceramic Society 1993, 76, 2707-2726
    •  
    • Page 7-8, Table1, Fig.6 : There is no comparison of calculated data with those published in the literature
    • Comparison of calculated data with those published in the literature by Kim [7], confirmed that ultrasonic spray pyrolysis can prepare also particle size of 100 nm using small concentration of solution of 0.1 g/L.
      • Page 9, line 2: Authors state: ‘’ …A decrease of solution concentration leads to smaller particle size as shown at figure 8.’’ However, particle size dependence on solution concentration is not shown on Fig.8
      • A decrease of solution concentration from 0.5 mol/L to 0.125 mol/ leads to smaller particle size as shown at figures 7 (left) and 8 (left), respectively.’’ 
      • Page 9, lines 3-6: Please, explain the sentence “…EDS analysis has confirmed the presence of silicon and oxygen togehter with elements such as Al, Cu, C, which are used in the preparation of samples for characterisation, as shown at Figure 9.” We improved it.
      • EDS analysis has confirmed the presence of silicon and oxygen togehter with elements such as Al, Cu, C, which are used as  carriers in the preparation of samples for characterisation, as shown at Figure 9.”

        These elements are not connected with our ultrasonic spray pyrolysis synthesis. Additionally, ICP-OES analysis of Si in solution before and after USP- Synthesis was included in our consideration. The concentration of Si was decreased from 13600 mg/L to 74.6 mg/l for 0.5 M precursor solution. Similar behavior was revailed for 0.125 M solution, where the concentration of Si was decreased from  3750 mg/l to 1 mg/l, what confirms fully transformation of the used precursor to SiO2.

      • Page 9, line 20: Looks like the term “Theoretical particle size“ means calculated particle size in the sentence “…..Theoretical particle size are situated in frame of maximal and minimal values of measured diameters, as shown in Table 2’. Please correct and edit English
      • We changed it: Calculated particle sizes are situated between maximal and minimal values of measured diameters, as shown in Table 2
  1. Conclusion needs to be corrected and detailed to present some real data obtained by authors

    Synthesis of silica powder was performed from a high concentrated colloidal solution (30 %) at 900 °C using ultrasonic spray pyrolysis method. This method enables a production of very fine spherical silica particles from irregular structure in one horizontal reactor. The controlled synthesis of particles was reached changing the concentration of precursor solution from 0.5 to 0.125 mol/L. A decrease of concentration from 0.5 to 0.125 mol/L leads to a decrases of measured average diameter from 690 nm to 610 nm. Hill like peak in the range of [2Ө]= 21-24 obtained by XRD-analysis, indicates the absence of any ordered crystalline structure and highly disordered structure of silica with high purity. Comparison of theoretically and measured diameter values of prepared silica has shown relatively good agreement, where a deviation amounts 17 % for average diameter. Disadvantage of this method is a collision of droplets during its transport using carrier gas, and especially low efficiency, due to losses of the dissolved precursor on the construction elements of reactor.

    Because of your excellent comments we added firstly 10 new references in order to explain better our results. We have invested many efforts in order to give good answers for your excellent remarks. I hope that this new Version is now suitable for Publishing in Metals. We improved also our references.

Round 2

Reviewer 2 Report

The manuscript has been revised extensively and now it can be accepted for publication in Metals. Congratulations!!

Reviewer 3 Report

I have no remarks